# Spin selection in atomic-level chiral metal oxide for photocatalysis

Minhua Ai[1,2], Lun Pan [1,2,3] ✉, Chengxiang Shi[1,2,3], Zhen-Feng Huang [1,2,3], Xiangwen Zhang[1,2,3], Wenbo Mi [4] & Ji-Jun Zou [1,2,3] ✉

The spin degree of freedom is an important and intrinsic parameter in boosting carrier dynamics and surface reaction kinetics of photocatalysis. Here we show that chiral structure in ZnO can induce spin selectivity effect to promote photocatalytic performance. The ZnO crystals synthesized using chiral methionine molecules as symmetry-breaking agents show hierarchical chirality. Magnetic circular dichroism spectroscopic and magnetic conductive-probe atomic force microscopic measurements demonstrate that chiral structure acts as spin filters and induces spin polarization in photoinduced carriers. The polarized carriers not only possess the prolonged carrier lifetime, but also increase the triplet species instead of singlet byproducts during reaction. Accordingly, the left- and right-hand chiral ZnO exhibit 2.0- and 1.9-times higher activity in photocatalytic $O_2$ production and 2.5- and 2.0-times higher activities in contaminant photodegradation, respectively, compared with achiral ZnO. This work provides a feasible strategy to manipulate the spin properties in metal oxides for electron spin-related redox catalysis.

Photosynthesis is the most efficient bio-system for converting solar energy into chemical energy. In the green plants' system, photons are absorbed by Mg-porphyrin complex P680 and P700, then water oxidation is catalyzed by $CaMn_4O_5$ clusters in photosystem II to produce $O_2$, while the excited electrons migrate along the high-speed transport chain to proton reduction sites in photosystem I[1,2] (Fig. 1a). These processes involve several spin-dependent steps, and the electron spin plays an indispensable role. First, photosystem II works as a spin filter for oxygen evolution[3], where the $Mn_4$ subsites with high-spin states in $CaMn_4O_5$ solely extract electrons with the same spin orientation from water and produces spin-aligned intermediate radicals (•OH). This energetically and entropically accelerates the formation of triplet-state $O_2$ and inhibits the formation of singlet species like $H_2O_2$ (Fig. 1b). Second, the electron transfer in PSI is highly spin-dependent because the electron transfer should satisfy spin angular momentum conservation, Hund's law and Pauli exclusion principle, all of which are related to the spin orientation of electrons. In addition, the spin of

migrating electrons is aligned parallelly to its linear momentum, which induces a high electron-transfer efficiency owing to the suppressed electron backscattering effect[4–8]. Numerous studies on photo(electro) catalysis, regarded as "artificial photosynthesis", have been carried out for the utilization and storage of solar energy[9] (Fig. 1c). However, the inefficient separation and migration of carriers and sluggish water oxidation still restrict the efficiency. This emphasizes the necessity of manipulating the spin-dependent properties of photocatalysts.

Theoretical and experimental works have been devoted to regulating the spin configuration and spin-dependent electron transfer of catalysts[10–14]. The spin configuration and orbital interaction of electrocatalysts (like $Ni_xFe_{1-x}OOH$ for oxygen evolution reaction and $FeN_4$ for oxygen reduction reaction) have been highlighted as pivotal factors[15–17]. Constructing magnetic ordering structures[18–20] and/or applying an external magnetic field[21–23] could build a spin channel to generate the spin-polarized electrons for optimizing the electron transfer at the interface between electrocatalysts and reactants/

[1]Key Laboratory for Green Chemical Technology of the Ministry of Education, School of Chemical Engineering and Technology, Tianjin University, Tianjin 300072, China. [2]Collaborative Innovative Center of Chemical Science and Engineering (Tianjin), Tianjin 300072, China. [3]Haihe Laboratory of Sustainable Chemical Transformations, Tianjin 300192, China. [4]Tianjin Key Laboratory of Low Dimensional Materials Physics and Preparation Technology, School of Science, Tianjin University, Tianjin 300354, China. ✉e-mail: panlun76@tju.edu.cn; jj_zou@tju.edu.cn

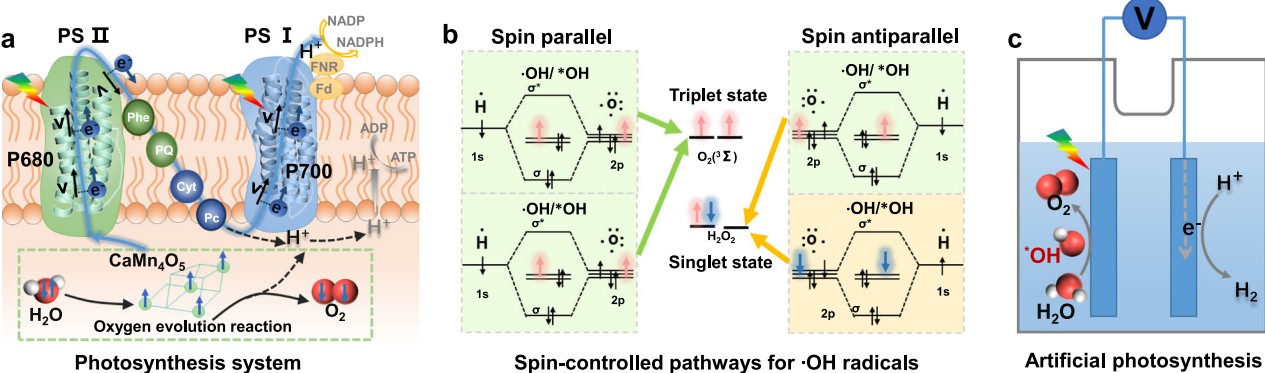

**Fig. 1 | The indispensable role of electron spin in the photosynthesis system. a** Scheme of oxygen evolution reaction and electron-transfer pathways in natural photosynthesis system. **b** Formation of singlet or triplet products by •OH radical with different spin directions. **c** Scheme of artificial photosynthesis.

intermediates. Also, a few spin-related researches have been carried out in photocatalysis[24–26]. Our group[27] manipulated highly spin-polarized electrons in Ti-defected $TiO_2$ and confirmed that the spatial spin polarization can efficiently promote the charge separation and surface reaction, which can be further strengthened by external magnetic fields. Jiang et al.[28] regulated the cobalt spin state by changing its oxidation state in COF-367-$Co^{II/III}$ to achieve excellent activity and selectivity in photocatalytic $CO_2$ reduction. Tsang et al.[29,30] investigated the magnetic-field-promoted photocatalytic overall water splitting and obtained an excellent solar-to-hydrogen efficiency, which is closely related to the prolonged exciton lifetime of $Fe_3O_4$/N-$TiO_2$ photocatalyst by Lorentz effect and spin-polarized effect. Basically, these efforts mainly rely on fine-tuning coordination structures and valence states of transition metals to regulate the spin degree of freedom.

A more universal approach to produce spin-polarized carriers is using spin filters, like chiral structures, like biomolecules (proteins, oligopeptides DNA and so on) assemblies[31–35] and especially chiral inorganic materials[36,37], termed as chiral-induced spin selectivity (CISS) effect[38–40]. Some exciting experimental results involving the CISS effect in chiral molecules have been reported, such as spin-selective transport, long-range spin electron transport, chiro-optical response and charge polarization, enantiospecific adsorption, and so on[38]. But it is still difficult to establish a quantitative relationship between optical response and the magnitude of the CISS effect. In the CISS effect, as the electron's velocity direction changes, the spin direction is also altered, which is equal to an effective magnetic field generated by the joint action of chiral structure and electric dipole moments. Subsequently, it can control the electron's spin direction and breaks the degeneracy of the spin energy level in the chiral structure. The effective magnetic field can be estimated by the motion law of charge in a magnetic field. Suppose that an electron of mass $m$ moves around in a chiral molecule with helix radius $r$, then $r = m\nu/e|B_{eff}|$, where e stands for electron charge and $\nu$ is for electron velocity. This can significantly affect spin-dependent electron transfer during catalysis.

Recently, chiral systems have been constructed by the assembly of chiral organic molecules on semiconductors like $TiO_2$[31–33] and $Fe_3O_4$[35] photoelectrodes, which can accelerate the photocatalytic oxygen evolution from water. However, the systems involving chiral macromolecules may bring some problems, such as instability when exposed to light and electric field, high conductivity barrier, and partially shielded active sites[39]. The direct construction of chiral oxides and their usage in photocatalysis should be more attractive. However, there are very few reports as to chiral oxides[41], and the effects of the nano-scale chiral structure on the charge transfer and oxygen evolution reaction in photocatalysis have not been considered.

Herein, we fabricated the chiral ZnO photocatalyst as the prototype by asymmetric coordination between the chiral center of the methionine molecules and $Zn^{2+}$ ions. The chiral structure acts as the spin selection filter and induces spin polarization during the separation of photoinduced carriers. Then, we found that, under the conservation of spin angular momentum, the photoexcited carrier lifetime is extended by suppressing the carrier recombination, and the spin-dependent kinetics of photocatalytic reactions are promoted by inhibiting the singlet $H_2O_2$ formation. As a consequence, compared with achiral ZnO, left- and right-hand chiral ZnO show significantly higher photocatalytic activity in $O_2$ production and photodegradation of pollutants.

## Results

### Chiral structure in ZnO

Chiral ZnO was fabricated using amino acid-induced self-assembly method[42], as shown in Fig. S1 (Supporting Information, SI). Chiral L- and D-methionine, with mirror-image symmetric transmitted circular dichroism (TCD) spectra (Fig. S2, SI), were used as the chiral inducer for asymmetric coordination with $Zn^{2+}$ ions. Upon hydrothermal synthesis, zinc carbonate hydroxide hydrate (JCPDS no. 11-0287) was first deposited on the FTO substrate as evidenced by XRD patterns (Fig. S3, SI). After pyrolysis at 550 °C, L- or D-ZnO with the chirality was produced. The weight loss during pyrolysis (Fig. S4, SI) is close to the theoretical mass loss, indicating the organic ligands are completely removed. Meanwhile, the FT-IR absorption peaks of carbonate groups disappear, while the peak at 422 $cm^{-1}$ assigned to the Zn−O stretching mode is significantly enhanced (Fig. S5, SI). In addition, racemic DL-methionine was used as a coordination agent to obtain achiral DL-ZnO for comparison.

The XRD patterns of as-prepared ZnO samples match well with the standard hexagonal wurtzite phase of ZnO (JCPDS no. 36–1451) without any impurities (Fig. S6, SI). SEM images show that all ZnO is composed of curved ultrathin nanosheets (Fig. S7, SI) with a uniform film thickness of ca. 1.35 ± 0.3 μm. From TEM images (Figure S8a−c, SI), the crystal lattice fringe of 0.26 nm is corresponding to the (002) facet of ZnO (Fig. S8d−i, SI). Homogeneous distribution and the similar mole ratio of Zn/O atoms are evidenced by EDX elemental mapping (Fig. S9, SI). All ZnO samples show a similar surface area and pore structure distribution (Fig. S10, SI), and the mesoporous structure can enhance the light scattering/reflection and provide abundant active sites[43]. The XPS spectra of samples (Fig. S11a, SI) show Zn $2p_{1/2}$ and $2p_{3/2}$ peaks at 1046.03 and 1022.98 eV, respectively, corresponding to $Zn^{2+}$ species. The high-resolution O 1s spectra fitted in Fig. S11b (SI) can be indexed into two distinctive binding energy peaks at 530.5 and 532.1 eV, which belong to the lattice oxygen and adsorbed oxygen species, respectively[44,45]. From UV−visible absorption, M−S plots, and XPS VB

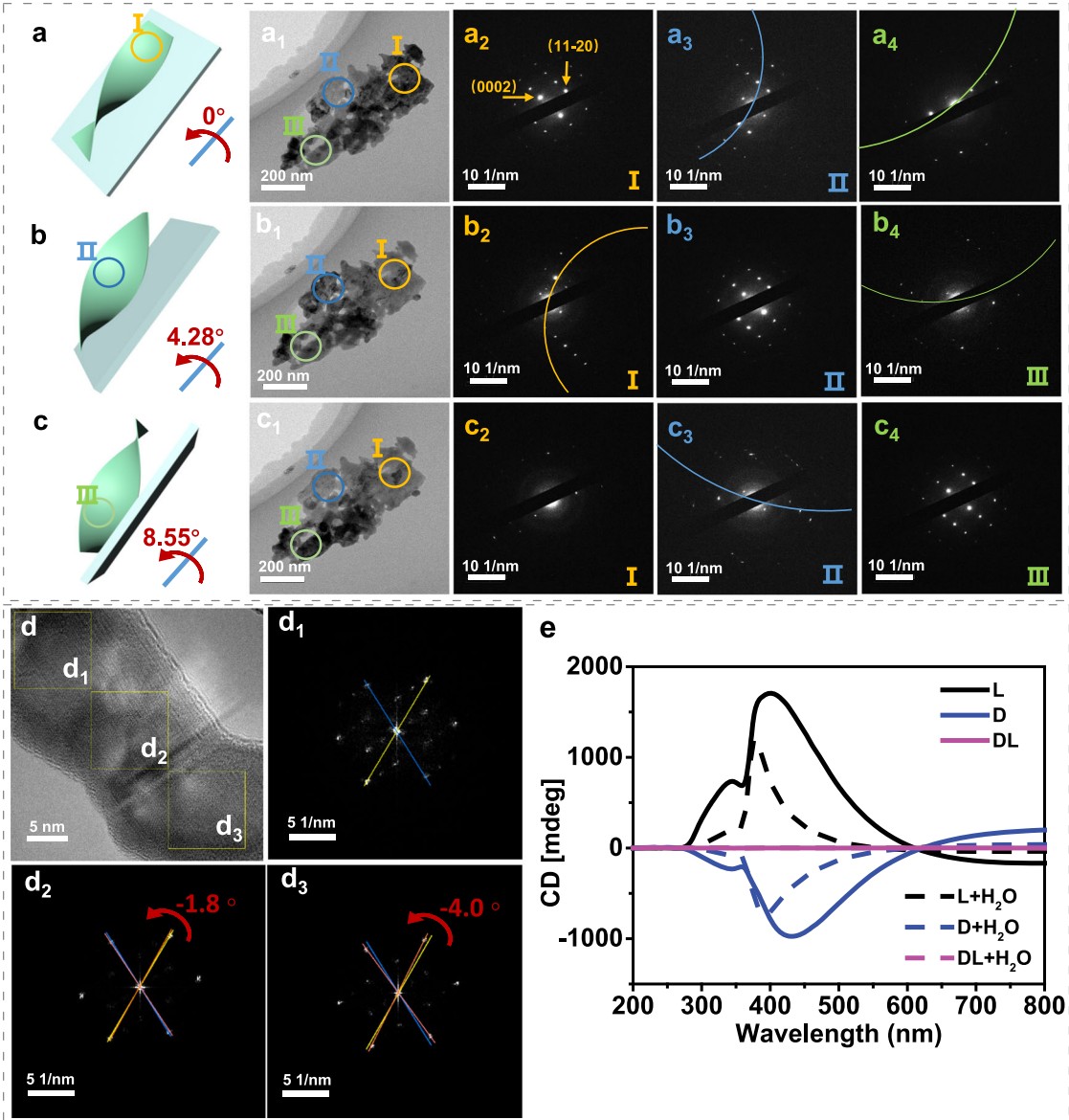

**Fig. 2 | Chiral structure of ZnO. a–c** Scheme of the structural model with different angles of 0°, 4.28°, and 8.55°, respectively. **a₁** TEM images at 0° and **a₂–a₄** corresponding SAED patterns of regions I–III of L-ZnO, respectively. **b₁** TEM images of the crystal tilted 4.28° along the axis [11−22] (The regions II aligned with an incident electron beam), and **b₂–b₄** corresponding SAED patterns of regions I–III of L-ZnO. **c₁** TEM images of the crystal tilted 8.55° along the axis [11−22] (The regions III aligned with incident electron beam), and **c₂–c₄** corresponding SAED patterns of L-ZnO. **d** HRTEM image and **d₁–d₃** Fast Fourier transform images of L-ZnO. **e** CD spectra of ZnO (solid lines) and after infiltration with water (dashed lines). Source data are provided as a Source Data file.

spectra (Figs. S12, SI), all samples show a similar absorption edge of 380 nm, corresponding to the bandgap of 3.08 eV, and similar flat potential and valence band positions. Overall, there are no significant differences in morphology, chemical composition, and band structure among chiral L/D-ZnO and achiral DL-ZnO.

The chiral structure of ZnO crystals was determined from atomic and nanostructure-assembled scales via TEM characterization. The selected-area electron diffraction (SAED) patterns obtained from low-resolution TEM images of L-ZnO confirm the chirality at the nanostructure-assembled level. In Fig. 2a, when the incident electron beam is aligned with the [−1100] axis, the periodic diffraction spots of ZnO crystal with a space group of P6₃mc can be observed in region I (Fig. 2a₁, a₂). Notably, the Laue zones (marked by dashed lines) appear in diffraction patterns of region II (Fig. 2a₃) and region III (Fig. 2a₄) because the crystal axis deviates from the direction of the incident electron beam[46]. The zone axes of region II in Fig. 2b and region III in Fig. 2c can be well aligned by tilting 4.28° and 8.55° along the [11−22]

axis, respectively. The inconsistent diffraction patterns in different regions under the same incident electron beam indicate L-ZnO crystals are assembled into helically bent nanoplates in a left-handed direction[42]. For atomic-level chiral structure, high-resolution TEM images are shown in Fig. 2d. Three consecutive fast Fourier transform (FFT) images of L-ZnO exhibit an anticlockwise rotation (ca. −4.0°) from the top (d₁) to the bottom (d₃) area at the atomic level chirality, while no related rotation happens for DL-ZnO (Fig. S13a, SI). Similarly, the diffraction spots of D-ZnO show a clockwise rotation of ca. +3° (Fig. S13b, SI). Based on the above results, the chirality of L-ZnO (or D-ZnO) originates from atomic level distortion and the assembled left-handed (or right-handed) chiral nanoplates, while DL-ZnO does not show any chiral structure.

The chiral structure can preferentially interact with left- or right-circularly polarized electromagnetic waves, which can be detected by TCD spectra[47]. In Fig. 2e, L-ZnO and D-ZnO show strong mirror-image TCD signals that are typical absorption characteristics of chiral

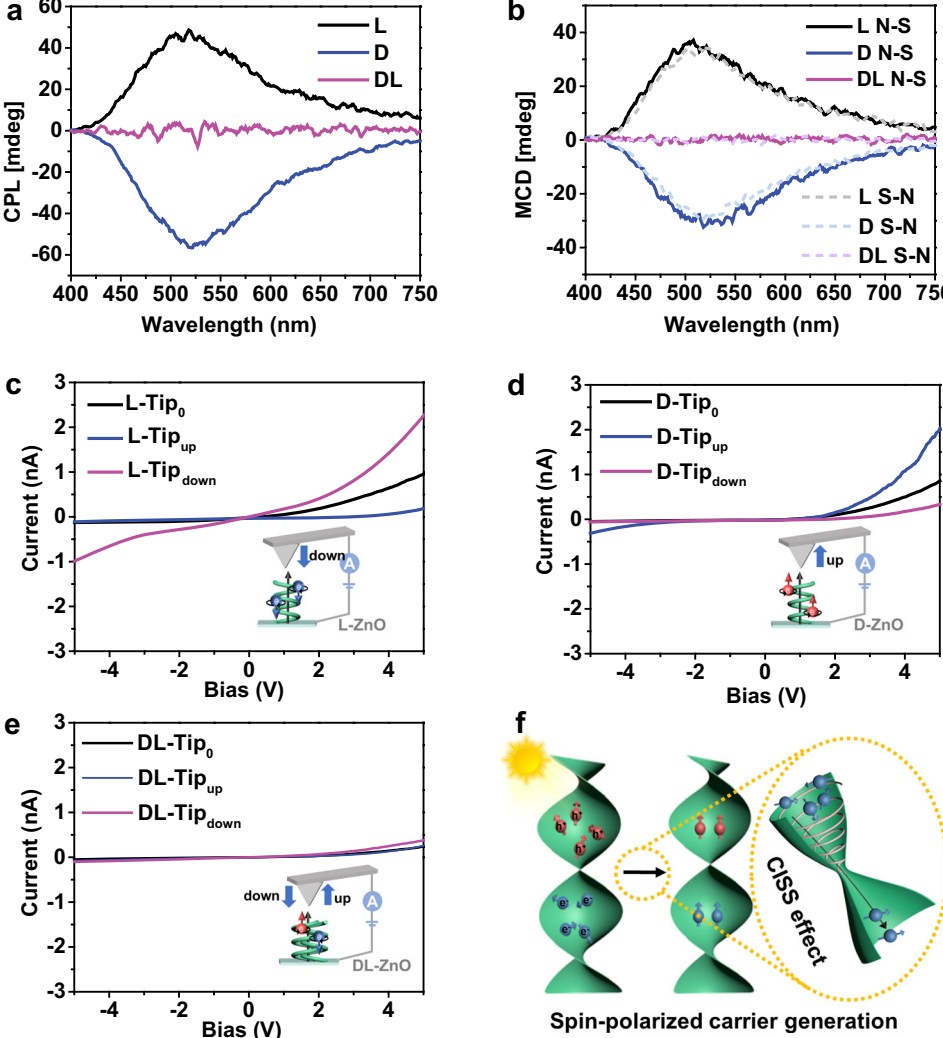

**Fig. 3 | Chirality-induced spin polarization of chiral ZnO. a** CPL spectra under 325 nm excitation. **b** MCD spectra measured under parallel (solid lines) and anti-parallel (dashed lines) magnetic fields. **c–e** Room-temperature *I–V* curves obtained by mc-AFM measurements. The tip is magnetized in up (blue), down (purple), and nonmagnetized (black) directions, respectively. **f** Schematic diagram of spin selection of photoinduced carriers in chiral ZnO. Source data are provided as a Source Data file.

structures, strongly confirming their chiral properties. For ʟ-ZnO, a high-intensity positive characteristic peak appears in the region of 300–600 nm and a weak negative signal in a long-wavelength region (>600 nm). Actually, the TCD spectra may include scattering- and absorption-based optical activities (OAs) (Fig. S14 SI)[48]. After infiltration with water (dashed lines in Fig. 2e) to eliminate the scattering-based OA, the intrinsic absorption-based OA is distinguished by subtracting the optical refractive index difference between the sample and water. In those cases, ʟ-ZnO and ᴅ-ZnO still show positive and negative absorption-based OA peak at ca. 380 nm, indicating that they prefer to absorb the left- and right-circularly polarized light (CPL), respectively. However, no chirality-related TCD signals appear for achiral ᴅʟ-ZnO.

**Spin selection in chiral ZnO**

The atomic-level chiral structure in oxides can generate a unique potential field during electron migration, which provides a possibility for the regulation of electron spin orientation[48–50]. Spin-polarized CPL emission tests were conducted, which can reflect OA in the excited states during the emission process while preserving spin angular momenta[51,52]. As shown in Fig. 3a, ʟ-ZnO with a left-handed chiral structure prefers to emit left-CPL and shows a positive CPL response, while ᴅ-ZnO with an antipodal structure shows the opposite signals.

These phenomena originate from the recombination between specific spin carriers[53]. Therefore, the spin-polarized electrons can be generated in chiral ZnO, which provides a platform for manipulating the spin-polarized states of photoexcited charges. On the contrary, achiral ᴅʟ-ZnO shows no spin-related CPL signals.

To further elucidate the spin selection in electron excitation, magnetic circular dichroism spectroscopy (MCD = $CD_{B\neq0} - CD_{B=0}$) was conducted owing to its particular sensitivity to spin-polarized states near the band-edge[48]. In ferromagnets, the electrons in the ground state can generate spin moments due to the spin−orbital coupling-induced spin splitting, which is sensitive to the external magnetic field. However, the chiral ZnO samples exhibit nonmagnetic behavior without saturation and hysteresis (Fig. S15, SI), which rules out the existence of spin moments caused by spin−orbital coupling-induced spin splitting. Figure 3b displays the MCD spectra under parallel and anti-parallel magnetic fields at room temperature. Chiral ʟ- and ᴅ-ZnO show positive and negative symmetric signals, respectively, which are not affected by changing the magnetic field direction (S−N or N−S), further confirming the chirality-induced spin selectivity (CISS) of photo-excited charges because it is a dynamic behavior. Specifically, for ʟ-ZnO, the left-handed circularly polarized light photoexcites spin-down electrons, forming spin-polarized excitons with a positive MCD signal,

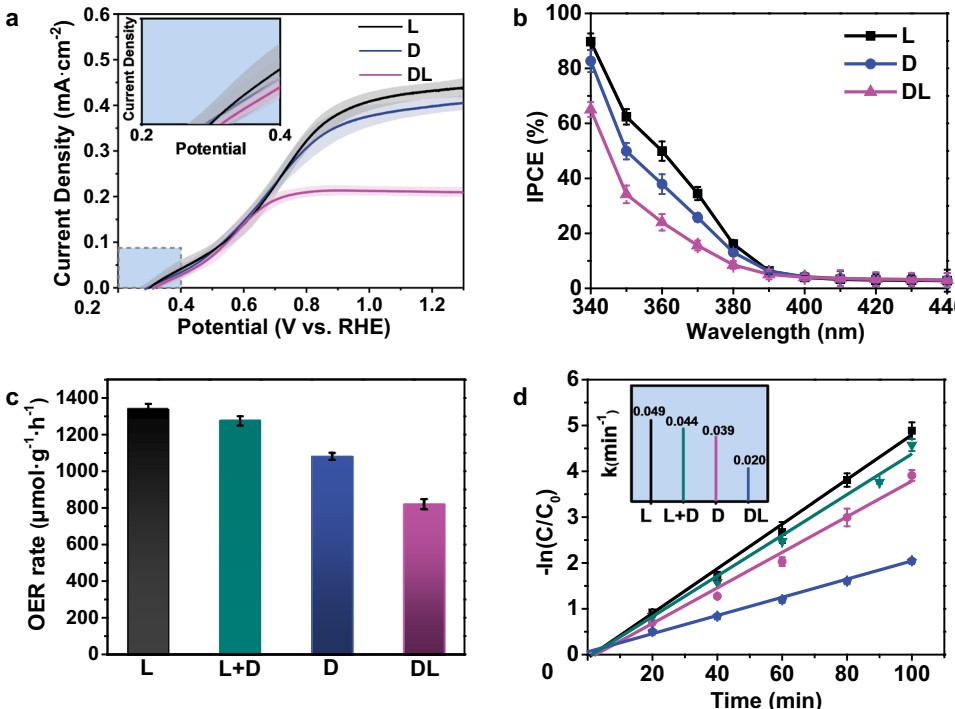

**Fig. 4 | Photocatalytic performance of chiral and achiral ZnO. a** LSV curves and **b** IPCE curves of photoanodes. **c** Photocatalytic oxygen production and **d** photodegradation rate of powdered photocatalysts (The inset is the calculated order rate constants ($k$)). L + D represents the physical mixture of L-ZnO and D-ZnO samples. The shaded areas around the data point indicate error bars. (Error bars are defined as s.d.) Source data are provided as a Source Data file.

while spin-up excitons are formed for right-handed D-ZnO. The signals maintain the same after the infiltration with water (Fig. S16, SI). No signals are observed for achiral DL-ZnO, further confirming the critical role of chiral structure on the spin selection of photoexcited charges.

Spin-selective charge transport in chiral ZnO can be evaluated via magnetic conductive-probe atomic force microscopy (mc-AFM). When a bias potential is applied, the transfer of electrons from the substrate to the tip can generate an electric current. The current–voltage ($I$–$V$) curves depending on the direction of tip magnetization (up or down relative to substrate), can reflect the spin polarity of the tunneling carriers[54]. The $I$–$V$ curves were obtained by averaging over 50 scanning data at room temperature (Fig. S17, SI). For L-ZnO, compared with the $I$–$V$ curve without specific magnetization direction (Tip$_0$), a higher tunneling current is recorded when the tip is in the down magnetization direction (Tip$_{down}$), while a lower current for that in the up direction (Tip$_{up}$) (Fig. 3c), suggesting that the electrons with a specific spin direction can easily transport in L-ZnO due to the CISS effect. However, the opposite charge transfer behavior appears for D-ZnO, where a higher current is detected when the tip is up-magnetized direction while the suppressed current occurs for that in the down direction (Fig. 3d). The degree of spin polarization $P$, defined as $P = (I_{up} - I_{down})/(I_{up} + I_{down}) \times 100\%$, is calculated from $I$–$V$ curves, which is −85% and +71.6% at 5 V for L- and D-ZnO, respectively. The high spin polarization indicates the excellent spin filter property in chiral L/D-ZnO catalysts. In contrast, achiral DL-ZnO does not show spin selectivity, showing an almost unchanged current (Fig. 3e). Thus, the photoexcited carriers with random spin directions in the chiral ZnO are modulated into spin-polarized ones attributed to the chiral potential field (Fig. 3f), which is expected to enhance both the carrier dynamics and surface reaction kinetics in photocatalysis[27,55,56].

### Spin-polarization-dependent photocatalysis

The photoelectrocatalytic water oxidation activity of ZnO under simulated solar light irradiation was then investigated. In Fig. 4a, the photocurrent density of achiral DL-ZnO photoanode at 1.23 V vs. RHE is 0.21 mA cm$^{-2}$, which is increased significantly to 0.43 and 0.40 mA cm$^{-2}$ for L- and D-ZnO with chiral structures, respectively. The reproducibility of PEC performance is also measured in Fig. S18, and the statistical onset potential of chiral ZnO is slightly lower than that of achiral ZnO (Inset in Fig. 4a). It is worth noting that L- and D-ZnO show high applied bias photon to current efficiency (0.175–0.19%) at 0.76 V vs. RHE, which is more than 1.6 times higher than that of DL-ZnO (0.11%), as shown in Fig. S19 (SI). Moreover, the incident photon-to-current conversion efficiency (IPCE) values of L- and D-ZnO photoanodes reach 36% and 27% at 370 nm, which are ca. 2.2 and 1.7 times higher than DL-ZnO (16%), respectively (Fig. 4b). In addition, the performance improvement is better than the reported modification strategies for ZnO-based photoanodes (Table S1, SI). As for the photocatalytic oxygen evolution with powder photocatalysts, the chiral L- and D-ZnO catalysts show an obviously higher O$_2$ generation rate than achiral DL-ZnO (Fig. 4c). For the photodegradation of Rhodamine B (RhB), the pseudo-first-order rate constants ($k$) are determined in Fig. 4d. The chiral L- and D-ZnO exhibit 2.5- and 2.0-times higher activity than DL-ZnO, respectively. Moreover, the physical mixture of L-ZnO and D-ZnO samples shows similar performance to individual L-ZnO and D-ZnO, indicating that spin selection occurs inside single particles. The above results confirm that the chiral photocatalyst shows much higher performance than the achiral one in both photoelectrochemical and photocatalytic oxidation reactions.

The high spin polarization induced by the directional potential field in the chiral structure can significantly prolong the carrier lifetime because the electron and hole recombination that should satisfy the conservation of spin angular momentum is inhibited[24,27,29]. The excited charge carrier dynamics of chiral and achiral ZnO were monitored by transient-absorption spectroscopic (TAS) measurement. Figure S20 (SI) and Fig. 5a–c, L-, D- and DL-ZnO show similar spectra, with the positive signals at a wavelength less than 360 nm and the negative peaks centered at 370 nm, which can be attributed to excited-state

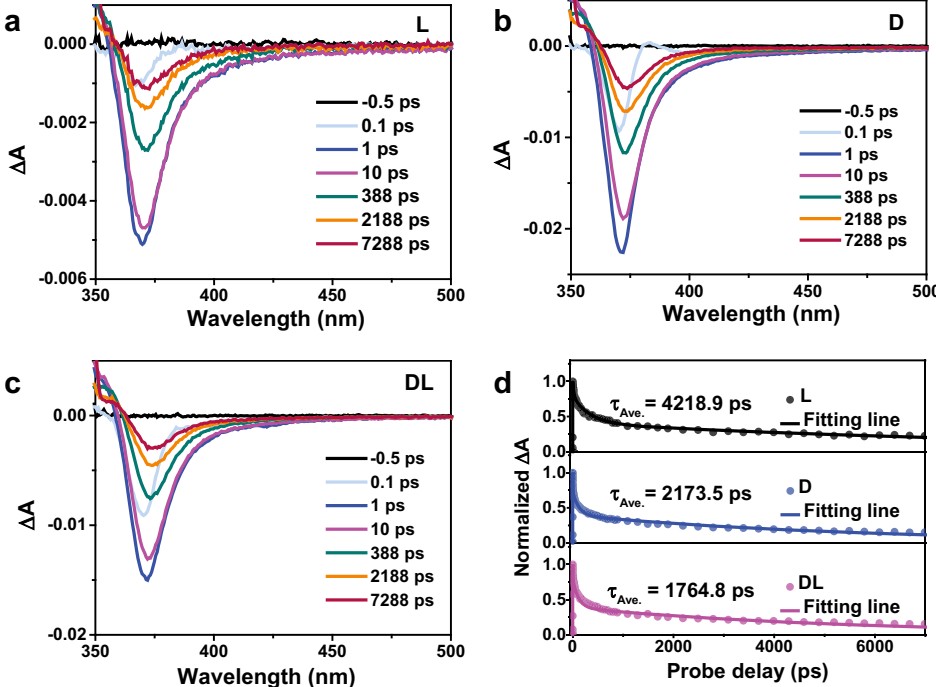

**Fig. 5 | Carrier dynamics in chiral and achiral ZnO.** The femtosecond time-resolved transient absorption spectra at several delay times of **a** L-ZnO, **b** D-ZnO, and **c** DL-ZnO. **d** Normalized carrier decay kinetics of L-, D-, and DL-ZnO. Source data are provided as a Source Data file.

absorption and ground state bleaching, respectively. The kinetics profiles are analyzed at the probing wavelength of 370 nm and fitted by a tri-exponential decay function accounting for electron trapping (Fig. 5d). Three decay processes for both chiral and achiral samples appear, corresponding to the trapping process ($\tau_1$), free carriers/exciton recombination ($\tau_2$), and trap state associated radiative/non-radiative recombination process ($\tau_3$) listed in Table S2 (SI). Importantly, the average carrier lifetimes of L- and D-ZnO are 4218.9 and 2173.5 ps, which are 2.4 and 1.2 times longer than DL-ZnO (1764.8 ps), respectively. Moreover, the contribution of the long-lifetime $\tau_3$ component, which is the dominant factor in charge transfer, increases from 31.3% of DL-ZnO to 40.6% and 37.3% of L- and D-ZnO, respectively.

Electron transfer between catalysts and reaction species should follow the Pauli exclusion principle and spin conservation principle at the reaction interface[8,57]. Herein, the spin polarization of photo-induced electrons correspondingly generates polarized holes, which will enhance reaction kinetics of triplet oxygen ($^3O_2$) formation in water oxidization and pollutants mineralization via suppressing the coupling of •OH active radicals to form $H_2O_2$[22,31,50,55]. The electron transfer number of the oxygen evolution reaction was investigated using rotating ring-disk electrode (RRDE) measurement (Fig. 6a). Compared with the electron transfer number of achiral DL-ZnO, a higher electron transfer number of L- and D-ZnO indicates that the formation of $H_2O_2$ is effectively inhibited, resulting in a higher pho-tocurrent and oxygen production rate. Furthermore, we measured the concentration of $H_2O_2$ during photoelectrochemical water oxidation with o-tolidine as an indicator. In Fig. 6b, there is an obvious peak at 436 nm (referring to $H_2O_2$) for DL-ZnO, indicating that two-electron water oxidation is the main competing process. While a ca. 5.5-fold reduced signal is detected for L- and D-ZnO, indicating the formation of $H_2O_2$ is inhibited to a large degree. The concentration of •OH active radicals generated in photocatalysis was also measured by in situ 5,5-dimethyl-1-pyrroline N-oxide (DMPO)-trapped EPR experiments (Fig. 6c). Both chiral and achiral samples show no characteristic EPR signals in the dark. Under light irradiation, the characteristic peaks of DMPO−•OH increase gradually with illumination time. After irradiating

for 8 min, the •OH peak intensity of chiral L- and D-ZnO is about two times higher than that of the achiral one, again confirming more •OH species are formed in the reaction system that finally induces higher photocatalytic performance. These results confirm the polarized holes significantly benefit the spin-dependent oxidization reactions (Fig. 6d).

## Discussion

This work modulates the spin polarization of ZnO crystals by tuning the atomic and bulk chiral structures to boost carrier dynamics and reaction kinetics of photocatalysis. Specifically, the chiral ZnO can create an effective magnetic field and serve as a spin filter via the CISS effect, which induces spin polarization of carriers. Importantly, the spin-polarized L- (or D-) ZnO shows a 2.4 (1.2) times longer charge carrier lifetime and ca. 5.5-fold reduced singlet byproducts than achiral ZnO. Accordingly, L- and D- ZnO samples exhibit 2.0- and 1.9-times higher activity in $O_2$ production and 2.5- and 2.0-times higher reaction rates in RhB degradation than achiral ZnO, respectively. Therefore, the construction of the inherent chiral structure in semiconductors pro-vides an effective way to induce spin polarization and thus provides a new perspective for improving carrier dynamics and surface reactions in photocatalysis. For this purpose, the development of a universal strategy to fabricate chiral photocatalysts is one key issue for fur-ther work.

## Methods
### Synthesis of chiral ZnO

Chiral ZnO samples were fabricated via an amino acid-induced self-assembly method with optimized parameters to get well ZnO photoanodes[42]. In a typical procedure, 2 mmol of L-methionine was added to 30 mL of ultrapure water and stirred until dissolved. Subse-quently, 3 mmol of zinc acetate was dissolved at 0 °C to form a uni-formly clarified solution. The activated FTO glass, according to the Supplementary Methods, was immersed, and 1 mmol ammonium carbonate was added and reacted under stirring for 30 min. The mix-ture and activated FTO glass were transferred to a Teflon-lined auto-clave (50 mL) for heating at 120 °C for 2 h. After cooling, the samples

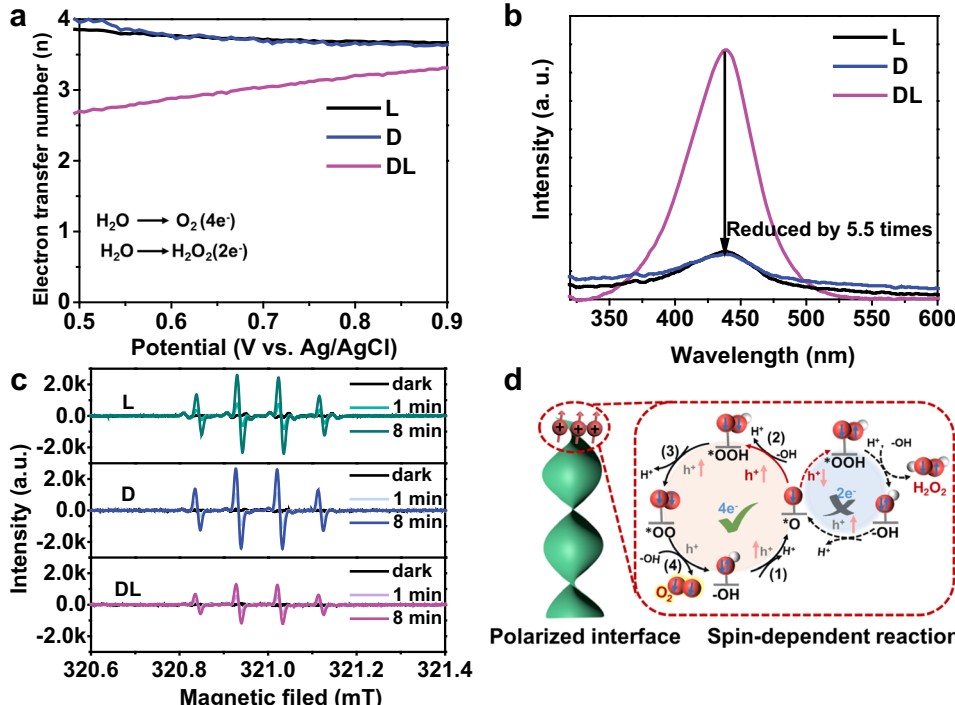

**Fig. 6 | Surface reaction over chiral and achiral ZnO. a** The calculated electron transfer number as a function of applied potentials via photoelectrochemical rotating ring-disk electrode measurement. **b** The amount of hydrogen peroxide with *o*-tolidine indicator detected by UV-vis absorption spectra. **c** EPR spectra of DMPO−•OH adducts in photocatalysis. **d** Schematic representation of spin-polarized surface reaction. Source data are provided as a Source Data file.

were cleaned with water and dried at 60 °C. Finally, chiral ZnO films were obtained under calcinating at 550 °C for 6 h. Films synthesized with L-, D-, and DL-methionine were defined as L-, D-, and DL-ZnO films, respectively. The corresponding ZnO powder samples were collected by centrifugation of precipitates from the reaction solution described above, followed by the same calcination treatment.

### Structure characterizations
The TCD spectra were conducted on a JASCO J-810 spectropolarimeter at room temperature. CPL spectra were measured by JASCO CPL-200 spectrophotometers with an external excitation source of 325 nm laser. MCD spectra were acquired using a JASCO J-1500 spectro-dichrometer and all the MCD spectra were probed under the magnetic field of ±1.6 T that was parallel or antiparallel with the light propagation direction and measured at the rate of 500 nm/min with a bandwidth of 10 nm. Multimode AFM with Nanoscope V controller (Bruker-Dimension icon) was used to record current–voltage ($I-V$) curves under a voltage bias of −6 to +6 V at the tip in a contact mode. $I-V$ curves were acquired using a magnetic Pt-coated Cr tip (Multi75E-G, Budget Sensors) with a nominal spring constant of 3 N/m, and the $I-V$ response was averaged over 50 scans for each sample. The tips are pre-magnetized in the up and down directions with a permanent magnet. Femtosecond TAS measurements were recorded by the Helios pump–probe system (Ultrafast Systems LLC). The optical parametric amplifier (TOPAS-800-fs) was adopted to provide the pump pulse of 340 nm (0.02 µj pulse$^{-1}$, 50 µJ cm$^{-2}$) at a typical focusing radius of ca. 150 µm. The irradiation area was 0.0004 cm$^2$. Other general structure characterizations are provided in the Supplementary Methods.

### Photoelectrochemical measurements
The photoelectrochemical measurements were performed using a CHI 660E electrochemical system (Shanghai, China) with a three-electrode system. A 300 W Xenon lamp (100 mW/cm$^2$, PLS-SXE300UV, Beijing Perfect Light Technology. Co. Ltd.) was equipped with an AM 1.5 filter

to simulate sunlight irradiation. Linear sweep voltammetry (LSV) curves were obtained at a scan rate of 10 mV/s in Na$_2$SO$_4$ (0.5 M) aqueous (pH = 6.8). Potentials were converted into reversible hydrogen electrodes (RHE) according to the equation: $E_{RHE} = E_{Ag/AgCl} + 0.059\,pH + 0.196$. In addition, at least 10 batches of samples were tested for photoanode performance, and error tapes were provided in Fig. 3a.

The RRDE measurement for the ZnO photoanode was performed on a Pine modulated speed rotator and an IVIUMSTAT workstation (Ivium Technologies BV). For RRDE measurements, 5 mg of ZnO powder that was scraped off from ZnO photoanodes was dispersed into 625 µL of water, 625 µL of ethanol, and 25 µL Nafion solution (5 wt %) under sonication for 60 min. Then, 10 µL of dispersion was drop-casted onto the glassy carbon disk electrode (surface area of *ca.* 0.196 cm$^2$) and dried at room temperature to form ZnO film. The ZnO disk/Pt ring current signals were collected in 0.5 M Na$_2$SO$_4$ in the dark and under irradiation at 1000 rpm.

### H$_2$O$_2$ detection
Hydrogen peroxide was detected by adding a redox indicator *o*-tolidine. The *o*-tolidine tends to be a yellow-colored solution in the presence of hydrogen peroxide, and the characteristic absorption peak is at 436 nm. The photoelectrochemical reaction was carried out for 3 h in a 0.5 M Na$_2$SO$_4$ (pH = 6.5) solution with a constant potential of 1.23 V vs. RHE. After that, 3 mL of the electrolyte solution was mixed with 1 mL of *o*-tolidine indicator solution, and the absorption spectra were monitored using a Shimadzu UV-2600 spectrometer.

### Photocatalytic tests
Photocatalytic oxygen evolution was evaluated in a closed Pyrex top-irradiation vessel (280 mL). Totally, 20 mg of catalyst decorated with CoO$_x$ cocatalyst was dispersed in H$_2$O (100 mL) with AgNO$_3$ (0.01 mol L$^{-1}$) as sacrifical agent. The reaction cell was purged with Argon for 30 min to remove air before irradiation. The light source

was a 300 W Xe lamp without any light filter, and the reaction temperature was kept at 15 °C. The produced $O_2$ was analyzed by gas chromatography (Bruker 450-GC) equipped with a thermal conductive detector.

Photocatalytic degradation of RhB was evaluated under xenon lamp irradiation. In a typical process, 20 mg ZnO nanoparticles were added to the 100 mL RhB (200 µmol/L) aqueous solution. The reaction solution was ultrasonicated for 15 min and magnetically stirred for 30 min in the dark to achieve the adsorption/desorption equilibrium. The temperature was kept at 25 °C. During the reaction, 1.5 mL reaction solution was withdrawn and centrifuged at 20-minute intervals. Then, the photocatalytic activities were analyzed with the UV–vis spectrophotometer (UV-2600, Shimadzu Ltd.). All photocatalytic experiments were repeated more than 3 times, and error bars were labeled.

## Data availability
All data needed to evaluate the conclusions in the paper are present in the paper and/or the Supplementary Information. Source data are provided as a Source Data file. Source data are provided in this paper.

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

## Acknowledgements

P.L., Z.J., and H.Z. appreciate the support from the National Natural Science Foundation of China (22222808, 22161142002, and 22121004) and the Haihe Laboratory of Sustainable Chemical Transformations for financial support.

## Author contributions

Z.J. and P.L. proposed the project and designed experiments. A.M. conceived and performed most of the experiments and characterizations. S.C. and H.Z. carried out the ESR characterizations and $H_2O_2$ detection. Z.X. and M.W. contributed to the analyses of CPL and MCD results. P.L. and A.M. wrote the paper. Z.J. and M.W. polished the writing of the paper. All authors discussed the results and commented on the paper.

## Competing interests

The authors declare no competing interests.
