## [Peer Review File · Nature Communications]

Spin selection in atomic-level chiral metal oxide for photocatalysisREVIEWER COMMENTS

Reviewer #1 (Remarks to the Author):

Zou and co-workers report the preparation of chiral ZnO nanocrystals that are employed in photocatalytic O₂ production. Combined experimental techniques are utilized to probe the chirality-induced spin-selectivity in O₂ production. The authors have described the work well, with sincere measurement efforts. However, considering the high impact of Nat. Commun. The present work lacks novelty which does not well fit the journal's high demand. My specific points are as follows:

1. A similar kind of work with the same chiral material has already been reported with ZnO, but photocatalytic oxidation was not performed; however, several researchers have already explored the spin-filtering effect through chiral materials by Ron groups and others. Thus, I don't think that the present work has sufficient novelty or newness.
2. In Fig 2d, compared to L-ZnO, Why in D-ZnO current drastically increase at the low potential in the Up-tip direction? Though spin-filtering is already well-studied in chiral materials, authors must briefly illustrate the mechanism in their systems.
3. If chiral ZnO shows enhanced electrocatalytic water oxidation activity compared to achiral ZnO, why is more potential required for water oxidation in chiral ZnO, as shown in Fig 3a?
4. The conclusion seems exactly the same as the abstract, which should be rewritten.
5. English/grammar is not up to the mark.
6. Challenges and future scope of this field are missing.

Reviewer #2 (Remarks to the Author):

The work described the preparation and characterization of chiral ZnO layers. Beyond serving as spin filter that can be used in reactions that involve triplet oxygen, the authors found that the chirality of the film prolongs the lifetime of the electron-hole following photoexcitation. The study is very interesting and convincing. However, two issues must be improved before this manuscript can be accepted for publication.

1. There are no error-bars in the data presented and not estimation of the accuracy of the result is given. In addition it is not clear how many samples were prepared and what is the variation in their properties.
 2. In the introduction it is not clear what of the properties of chiral systems mentioned were really observed and what properties one can assume to exist but they were not observed so far. A clear distinguishing between the two class must be made.
- Once the authors correct that two points the manuscript should be published in Nat. Comm.

Reviewer #3 (Remarks to the Author):

Most researchers regulated the spin degree of freedom in transition metals directly, which depends greatly on fine-tuning the coordinated structure of metals or producing spin-polarized carriers by using spin filters. In this study, the authors fabricated the hierarchical chirality of ZnO photocatalyst as the prototype by asymmetric coordination between the chiral center of the methionine molecules and Zn²⁺ ions. The chiral structures acted as the spin selection filter and induced spin polarization during the photoinduced carrier separation. Interestingly, the photoexcited carrier lifetime was extended by suppressing the carrier recombination, and spin-dependent kinetics of water oxidation reaction and pollutants mineralization reaction was significantly promoted by inhibiting the singlet H₂O₂ formation, and as a result, the left- and right-hand chiral ZnO showed greatly enhanced photocatalytic activity for the O₂ production and photodegradation of pollutants. Overall, it is an interesting

study, and the conclusions are mostly supported by the data, which may merit publication after fully addressing the following issues.

- 1) Page number is missing in the manuscript.
- 2) Abstract: “respectively” should be added immediately after “photodegradation”
- 3) Introduction: some closely-related publications need to be cited to enhance the research background, e.g, magnetic-field-promoted photocatalytic overall water-splitting (Energy Environ. Sci., 2022, 15, 265-277; Chem Catal. 2022, 2, 221-241)
- 4) The authors claimed that chiral ZnO was fabricated by adopting the amino acid-induced self-assembly method reported in the literature (ref. 40). Did the authors modify the method?
- 5) Figure S2, caption: “is precursor” should be replaced by “are the precursor”
- 6) Figure S7h: the crystal lattice fringe should be $2.64 \text{ nm} / 9 = 0.2933 \text{ nm}$ rather than ca. 0.26 nm. Please double-check.
- 7) Figure S8b-b2: why a different scale (250 nm) was used for these figures? It would be better to use the same scale as shown in S8a and S8c (i.e., 200 nm)
- 8) Figure S9: pore size distribution should be also provided in this figure.
- 9) Are there any mesopores or hierarchical porosity in the as-synthesized materials? 10) The authors mentioned that “all of the fabricated ZnO samples show a similar surface area (Figure S9, SI),” But they did not tell what the pore size is for the as-synthesized porous materials. In literature, the presence of mesopores (or macropores) favors multilight scattering/reflection, resulting in enhanced harvesting of the exciting light and thus improved photocatalytic activity (CrystEngCommun 2022, 24, 6498-6504; J. Clean. Prod. 2021, 328, 129745; Environ. Chem. Lett. 2021, 19, 3573-3582). The authors are suggested to consider similar effects from the mesopores (or macropores) if applicable.
- 11) The authors stated that “the high-resolution O 1s spectra in Figure S10b (SI) can be indexed into two distinctive binding energy peaks at 529.9 and 531.8 eV, which belong to the lattice oxygen and adsorbed oxygen/hydroxyl radicals on the surface, respectively”. Please provide literature here.
- 12) Reference 14: volume number (32) is missing. In addition, the article number is 2003297 rather than e2003297.
- 13) Methods: please change “1 mL o-tolidine indicator solution” to “1 mL of o-tolidine indicator solution”; change “20 mg catalyst decorated with CoOx cocatalyst” to “20 mg of catalyst decorated with CoOx cocatalyst”; change “0.01 mol·L⁻¹” to “0.01 mol L⁻¹”. Please correct similar errors throughout the manuscript.
- 14) Reference 17: article number is 1907976 rather than e1907976.
- 15) Reference 23: volume number (6) is missing.
- 16) It would be better to compare the photocatalytic performance of the as-synthesized photocatalysts in this study with that reported in the literature to justify that this study significantly advances the development in this field and thus merits publication in Nature Communications.

Dear Editor & Reviewers,

We would like to thank you for the constructive comments and suggestions, which are very helpful for us to improve our work. We have revised the manuscript carefully, and the followings are the point-to-point responses and list of modifications.

Reviewer 1

Zou and co-workers report the preparation of chiral ZnO nanocrystals that are employed in photocatalytic O₂ production. Combined experimental techniques are utilized to probe the chirality-induced spin-selectivity in O₂ production. The authors have described the work well, with sincere measurement efforts. However, considering the high impact of Nat. Commun. The present work lacks novelty which does not well fit the journal's high demand.

Comment 1:

A similar kind of work with the same chiral material has already been reported with ZnO, but photocatalytic oxidation was not performed; however, several researchers have already explored the spin-filtering effect through chiral materials by Ron groups and others. Thus, I don't think that the present work has sufficient novelty or newness.

Response 1:

Thanks for the very valuable comment. Chiral materials and their unique properties are attracting a growing interest in fields of materials and catalysis. The previous works are very enlightening and provide the solid basis for our work. To the authors' knowledge, however, there are only a few reports for photocatalysis using chiral properties of photocatalysts, such as TiO₂ photoanode (*J. Phys. Chem. Lett.* 2015, 6: 4916-4922; *J. Am. Chem. Soc.* 2017, 139: 2794-2798; *J. Phys. Chem. C* 2017, 121: 15777-15783) and Fe₃O₄ (*ACS Energy Lett.* 2018, 3: 2308-2313) assembled with chiral macromolecules (like proteins and DNA molecules).

As can be seen, most of the previous studies were limited to the use of chiral macromolecules to modify the surface of traditional photocatalysts. Some problems may be overlooked in those systems, such as instability, additional conductivity

barrier and limited active sites upon the introduction of organic macromolecules. The direct construction of chiral oxides and usage in photocatalysis should be more attractive.

Although chiral ZnO has been prepared and multiple optical activities such as high circularly polarized luminescence and Raman optical activity have been demonstrated in the literature, the CISS effect in this material and the specific mechanism in photocatalysis has not been considered. In this work, we for the first time report how chiral metal oxides improve the carrier transport and enhance the surface redox reaction. The result demonstrates the promising potential of chiral control in photocatalysis, and the chiral metal oxide is expected to represent a new direction of photocatalyst and attracts wide interests and follow-up study. We believe our work is a big progress in photocatalysis.

Modification 1:

1. The introduction was revised as follows.

A more universal approach to produce spin polarized carriers is using spin filters, like chiral structures, like biomolecules (proteins, oligopeptides DNA and so on) assemblies³¹⁻³⁵ and especially chiral inorganic materials^{36,37}, termed as chiral-induced spin selectivity (CISS) effect³⁸⁻⁴⁰. Some exciting experimental results involving the CISS effect in chiral molecules have been reported, such as spin-selective transport, long-range spin electron transport, chiro-optical response and charge polarization, enantiospecific adsorption and so on³⁸. But it is still difficult to establish a quantitative relationship between optical response and the magnitude of CISS effect. In CISS effect, as the electron's velocity direction changes, the spin direction is also altered, which is equal to an effective magnetic field generated by the joint action of chiral structure and electric dipole moments. Subsequently it can control the electron's spin direction and breaks the degeneracy of spin energy level in chiral structure. The effective magnetic field can be estimated by the motion law of charge in magnetic field. Suppose that an electron of mass m moves around in a chiral molecule with helix radius r , then $r = mv/e|B_{\text{eff}}|$, where e stands for electron charge and v is for electron velocity. This can significantly affect the spin-dependent electron transfer

during the catalysis.

Recently, the chiral systems have been constructed by assembly of chiral organic molecules on semiconductor like TiO_2 ³¹⁻³³ and Fe_3O_4 ³⁵ photoelectrodes, which can accelerate the photocatalytic oxygen evolution from water. However, the systems involving chiral macromolecules may bring some problems, such as the instability when exposing to light and electric field, high conductivity barrier and partially shielded active sites³⁹. The direct construction of chiral oxides and usage in photocatalysis should be more attractive. However, there are very less reports as to chiral oxides⁴¹, and the effects of the nano-scale chiral structure on the charge transfer and oxygen evolution reaction in photocatalysis has not been considered.

2. The following references were supplemented.

41. Vadakkayil A, Clever C, Kunzler KN, Tan S, Bloom BP, Waldeck DH. Chiral electrocatalysts eclipse water splitting metrics through spin control. *Nat. Commun.* **14**, 1067 (2023).

Comment 2:

In Fig 2d, compared to L-ZnO, why in D-ZnO current drastically increase at the low potential in the up-tip direction? Though spin-filtering is already well-studied in chiral materials, authors must briefly illustrate the mechanism in their systems.

Response 2:

Thanks for the valuable comment. The drastic current increase for D-ZnO at the low potential in the up-tip direction should be caused by the too narrow area scan of the surface of D-ZnO film, and these local structures may show relatively higher conductivity. To increase the testing accuracy, we re-tested the data for D-ZnO by increasing the selected testing points and area of the sample, and update the data in the revised manuscript. As shown in revised Figure 2d, the spin-dependent current increase for D-ZnO (D-Tip_{up}) is similar to L-ZnO (L-Tip_{down}). Moreover, the original data were also added in Figure S17 (Supporting Information). We also introduce the mechanism briefly in the manuscript.

Modification 2:

1. Page 10, Lines 3-9, the following sentences were modified:

Spin-selective charge transport in chiral ZnO can be evaluated *via* the magnetic conductive-probe atomic force microscopy (mc-AFM). When a bias potential is applied, the transfer of electrons from the substrate to the tip can generate electric current. The current-voltage (I-V) curves depending on the direction of tip magnetization (up or down relative to substrate) can reflect the spin polarity of the tunneling carriers⁵⁴. The I-V curves were obtained by averaging over 50 scanning data at room temperature (Figure S17, SI).

2. Figure 2d was revised.

Figure 2. Chirality-induced spin polarization of chiral ZnO. (a) CPL spectra under 325 nm excitation. (b) MCD spectra measured under parallel (solid lines) and

antiparallel (dashed lines) magnetic fields. (c-e) Room-temperature I - V curves obtained by mc-AFM measurements. The tip is magnetized in up (blue line), down (purple line), and nonmagnetized (black line) directions, respectively. (f) Schematic diagram of spin selection of photoinduced carriers in chiral ZnO.

3. The following references were supplemented.

54. Lu, H. P. *et al.* Spin-dependent charge transport through 2D chiral hybrid lead-iodide perovskites. *Sci. Adv.* **5**, eaay0571 (2019).

4. Figure S17 was revised.

Figure S17. I - V curves obtained from L-ZnO, D-ZnO and DL-ZnO using mc-AFM. (d-f) The current as a function of the applied voltage I - V curves of D-ZnO with the tips magnetized in the nonmagnetized (black), up (blue) and down (purple).

Comment 3:

If chiral ZnO shows enhanced electrocatalytic water oxidation activity compared to achiral ZnO, why is more potential required for water oxidation in chiral ZnO, as shown in Fig 3a?

Response 3:

Thanks for the very valuable comment. Onset potential and photocurrent are two fundamental parameters for evaluating the performance of photoelectrode. The potential at which OER begins under light is defined as the onset potential (E_{onset}). Meanwhile, the photocurrent density at 1.23 V_{RHE} is the most important benchmark for photoelectrochemical performance, since it is the theoretical potential where the oxygen evolution reaction can occur. In the original Figure 3a, the photocurrent density of achiral DL-ZnO photoanode at 1.23 V vs. RHE is $0.21 \text{ mA}\cdot\text{cm}^{-2}$, which is

increased significantly to 0.43 and 0.40 $\text{mA}\cdot\text{cm}^{-2}$ for L- and D-ZnO with chiral structures, respectively. This indicates the chiral ZnO has much higher catalytic activity. However, the onset potential of chiral ZnO is slightly higher than the achiral one, which is unnormal. To double check this, we re-tested the photoelectrochemical performances of 10 batches of L-, D-, and DL-ZnO samples as shown in Figure S18a-c (Supporting Information), and provided the averaged data in Figure 3a and Figure S18d (Supporting Information). The results confirm that L- and D-ZnO show relatively lower E_{onset} than DL-ZnO, further confirming the chiral ZnO is more active than achiral one.

Modification 3:

1. Figure S18 (SI) was revised as follows.

Figure S18. Reproducibility of PEC performance of (a) L-ZnO, (b) D-ZnO and (c) DL-ZnO. (d) Statistics of onset potential and (e) current density at 1.23 V vs. RHE of L-ZnO, D-ZnO and DL-ZnO.

2. Figure 3a was revised as follows.

Figure 3. Photocatalytic performance of chiral and achiral ZnO (The reproducibility of PEC performance is shown in Figure S18, SI). (a) LSV curves. (b) IPCE curves. (c) Photocatalytic oxygen production rate. (d) Photodegradation rate.

3. Pages 10-11, the following sentences were modified.

In **Figure 3a**, the photocurrent density of achiral DL-ZnO photoanode at 1.23 V vs. RHE is 0.21 mA·cm⁻², which is increased significantly to 0.43 and 0.40 mA·cm⁻² for L- and D-ZnO with chiral structures, respectively. The reproducibility of PEC performance is also measured in the Figure S18, and the statistical onset potential of chiral ZnO is slightly lower than that of achiral ZnO (Inset in Figure 3a). It is worth noting that, L- and D-ZnO show high applied bias photon to current efficiency (0.175%-0.19 %) at 0.76 V vs. RHE, which is more than 1.6 times higher than that of DL-ZnO (0.11%), as shown in Figure S19 (SI). Moreover, the incident photon-to-current conversion efficiency (IPCE) values of L- and D-ZnO photoanodes reach 36% and 27% at 370 nm, which are *ca.* 2.2 and 1.7 times higher than DL-ZnO (16%), respectively (**Figure 3b**). In addition, the performance improvement is better than the reported modification strategies for ZnO-based photoanodes (Table S1, SI).

Comment 4:

The conclusion seems exactly the same as the abstract, which should be rewritten.

Response 4:

Thanks for the very valuable comment. We have re-written both the abstract and conclusion.

Modification 4:

1. The abstract was revised as follows.

The spin degree of freedom is an important and intrinsic parameter in boosting the carrier dynamics and surface reaction kinetics of photocatalysis. Here, we demonstrate that chiral structure in ZnO can induce spin selectivity effect to promote photocatalytic performance. The ZnO crystals synthesized using chiral methionine molecules as symmetry-breaking agents show chirality in atomic and bulk levels. Magnetic circular dichroism spectroscopic and magnetic conductive-probe atomic force microscopic measurements demonstrate that chiral structure acts as spin filters and induces spin polarization in photoinduced carriers. The polarized carriers not only possess the prolonged lifetime during transfer process, but also increase the triplet active species instead of singlet byproducts during surface oxidation reaction. Accordingly, the left- and right-hand chiral ZnO exhibit 2.0- and 1.9-times higher activity in photocatalytic O₂ production and 2.5- and 2.0-times higher activities in RhB photodegradation, respectively, compared with achiral ZnO. This work provides a feasible strategy to manipulate the spin properties in metal oxides for electron-spin-related redox catalysis.

2. The conclusion was revised as follows.

This work modulates the spin polarization of ZnO crystals by tuning the atomic and bulk chiral structures, to boost carrier dynamics and reaction kinetics of photocatalysis. Specifically, the chiral ZnO can create effective magnetic field and serve as spin filters *via* CISS effect, which induces spin polarization of carriers. Importantly, the spin-polarized L- (or D-) ZnO shows 2.4 (1.2) times longer charge carrier lifetime and *ca.* 5.5-fold reduced singlet byproducts than achiral ZnO. Accordingly, L- and D- ZnO samples exhibit 2.0- and 1.9-times higher activity in O₂

production and 2.5- and 2.0-times higher reaction rate in RhB degradation than achiral ZnO, respectively. Therefore, the construction of inherent chiral structure in semiconductor provides an effective way to induce the spin polarization and thus provide a new perspective for improving carrier dynamics and surface reactions in photocatalysis. For this purpose, the development of universal strategy to fabricate chiral photocatalysts is one key issue for further work.

Comment 5:

English/grammar is not up to the mark.

Response & Modification 5:

Thanks for the very valuable comment. We checked the whole manuscript and carefully revised the English/grammar, with the help of native English speaker.

Comment 6:

Challenges and future scope of this field are missing.

Response 6:

Thanks for the very valuable suggestion. We have supplemented the challenges and future scope of this field in the conclusion section.

Modification 6:

The conclusion section was revised as follows:

This work modulates the spin polarization of ZnO crystals by tuning the atomic and bulk chiral structures, to boost carrier dynamics and reaction kinetics of photocatalysis. Specifically, the chiral ZnO can create effective magnetic field and serve as spin filters *via* CISS effect, which induces spin polarization of carriers. Importantly, the spin-polarized L- (or D-) ZnO shows 2.4 (1.2) times longer charge carrier lifetime and *ca.* 5.5-fold reduced singlet byproducts than achiral ZnO. Accordingly, L- and D- ZnO samples exhibit 2.0- and 1.9-times higher activity in O₂ production and 2.5- and 2.0-times higher reaction rate in RhB degradation than achiral ZnO, respectively. Therefore, the construction of inherent chiral structure in semiconductor provides an effective way to induce the spin polarization and thus

provide a new perspective for improving carrier dynamics and surface reactions in photocatalysis. For this purpose, the development of universal strategy to fabricate chiral photocatalysts is one key issue for further work.

Reviewer 2

The work described the preparation and characterization of chiral ZnO layers. Beyond serving as spin filter that can be used in reactions that involve triplet oxygen, the authors found that the chirality of the film prolongs the lifetime of the electron-hole following photoexcitation. The study is very interesting and convincing. However, two issues must be improved before this manuscript can be accepted for publication.

Comment 1:

There are no error-bars in the data presented and not estimation of the accuracy of the result is given. In addition, it is not clear how many samples were prepared and what is the variation in their properties.

Response 1:

Thanks for the very valuable suggestion. We have added detailed sample numbers for measurements and added the error bars in figures.

Modification 1:

1. **Figure S18** (SI) was supplemented as follows.

Figure S18. Reproducibility of PEC performance of (a) L-ZnO, (b) D-ZnO and (c) DL-ZnO. (d) Statistics of onset potential and (e) current density at 1.23 V vs. RHE of L-ZnO, D-ZnO and DL-ZnO.

2. **Figure 3a** was revised as follows.

Figure 3. Photocatalytic performance of chiral and achiral ZnO (The reproducibility of PEC performance is shown in Figure S18, SI). (a) LSV curves. (b) IPCE curves. (c) Photocatalytic oxygen production rate. (d) Photodegradation rate.

3. Page 10, Lines 26-31, the following sentences were modified.

In **Figure 3a**, the photocurrent density of achiral DL-ZnO photoanode at 1.23 V vs. RHE is 0.21 mA·cm⁻², which is increased significantly to 0.43 and 0.40 mA·cm⁻² for L- and D-ZnO with chiral structures, respectively. The reproducibility of PEC performance is also measured in the Figure S18, and the statistical onset potential of chiral ZnO is slightly lower than that of achiral ZnO (Inset in Figure 3a).

4. In Methods, Page 16, Lines 7-8 and Page 17, Line 4-5, the following sentences were added.

In addition, at least 10 batches of samples were tested for photoanode performance and error tapes were provided in Figure 3a.

All photocatalytic experiments were repeated more than 3 times and error bars were labeled.

Comment 2:

In the introduction it is not clear what of the properties of chiral systems mentioned were really observed and what properties one can assume to exist but they were not observed so far. A clear distinguishing between the two class must be made.

Response 2:

Thanks for the very valuable suggestion. We have added to the discussion of properties of chiral systems in the introduction section.

Modification 2:

1. Page 4, Lines 19-26, the following sentences were modified:

A more universal approach to produce spin polarized carriers is using spin filters, like chiral structures, like biomolecules (proteins, oligopeptides DNA and so on) assemblies³¹⁻³⁵ and especially chiral inorganic materials^{36, 37}, termed as chiral-induced spin selectivity (CISS) effect³⁸⁻⁴⁰. Some exciting experimental results involving the CISS effect in chiral molecules have been reported, such as spin-selective transport, long-range spin electron transport, chiro-optical response and charge polarization, enantiospecific adsorption and so on³⁸. But it is still difficult to establish a quantitative relationship between optical response and the magnitude of CISS effect.

Reviewer 3

Most researchers regulated the spin degree of freedom in transition metals directly, which depends greatly on fine-tuning the coordinated structure of metals or producing spin-polarized carriers by using spin filters. In this study, the authors fabricated the hierarchical chirality of ZnO photocatalyst as the prototype by asymmetric coordination between the chiral center of the methionine molecules and Zn²⁺ ions. The chiral structures acted as the spin selection filter and induced spin polarization during the photoinduced carrier separation. Interestingly, the photoexcited carrier lifetime was extended by suppressing the carrier recombination, and spin-dependent kinetics of water oxidation reaction and pollutants mineralization reaction was significantly promoted by inhibiting the singlet H₂O₂ formation, and as a result, the left- and right-hand chiral ZnO showed greatly enhanced photocatalytic activity for the O₂ production and photodegradation of pollutants. Overall, it is an interesting study, and the conclusions are mostly supported by the data, which may merit publication after fully addressing the following issues.

Comment 1:

Page number is missing in the manuscript.

Response & modification 1:

Thanks for the valuable suggestion. We have added page numbers in the revised manuscript.

Comment 2:

Abstract: “respectively” should be added immediately after “photodegradation”.

Response 2:

Thanks for the very valuable suggestion. We have added “respectively” after “photodegradation”.

Modification 2:

1. In Abstract, the following sentences were revised.

Accordingly, the left- and right-hand chiral ZnO exhibit 2.0- and 1.9-times higher activity in photocatalytic O₂ production and 2.5- and 2.0-times higher activities in

RhB photodegradation, respectively, compared with achiral ZnO.

Comment 3:

Introduction: some closely-related publications need to be cited to enhance the research background, *e.g.*, magnetic-field-promoted photocatalytic overall water-splitting (Energy Environ. Sci., 2022, 15, 265-277; Chem Catal. 2022, 2, 221-241).

Response 3:

Thank for the very valuable suggestion. We have added these important references in the introduction to enhance the research background.

Modification 3:

1. Page 4, Lines 13-16, the following sentences were added.

Tsang et al.^{29,30} investigated the magnetic-field-promoted photocatalytic overall water splitting and obtained an excellent solar-to-hydrogen efficiency, which is closely related to the prolonged exciton lifetime of Fe₃O₄/N-TiO₂ photocatalyst by Lorentz effect and spin-polarized effect.

2. The following references were supplemented.

29. Li, Y. *et al.* Local magnetic spin mismatch promoting photocatalytic overall water splitting with exceptional solar-to-hydrogen efficiency. *Energy Environ. Sci.* **15**, 265-277 (2022).

30. Li, C., Liao, G., Fang, B. Magnetic-field-promoted photocatalytic overall water-splitting systems. *Chem Catal.* **2**, 238-241 (2022).

Comment 4:

The authors claimed that chiral ZnO was fabricated by adopting the amino acid-induced self-assembly method reported in the literature (ref. 40). Did the authors modify the method?

Response 4:

Thanks for the very valuable suggestion. We have optimized the synthesis method for chiral ZnO. To fabricate the photoanode, we applied the fluorine-doped tin oxide

(FTO) glass as substrate to increase the conductivity. Moreover, we systematically optimized the synthetic parameters (*i.e.*, reagent dosage, reaction time, etc.) to improve the light adsorption of ZnO films and photoanode performance.

Modification 4:

1. Page 15, Lines 4-6, the following sentences were revised.

Chiral ZnO samples were fabricated *via* amino acid-induced self-assembly method according to the previous literature⁴² but with optimized the parameters to get well ZnO photoanodes.

Comment 5:

Figure S2, caption: “is precursor” should be replaced by “are the precursors”.

Response 5:

Thanks for the very valuable suggestion. We have replaced “is precursor” to “are the precursors”.

Modification 5:

1. Page 5 in Supplementary Information, the following sentences were revised.

Figure S2. X-ray diffraction patterns of zinc carbonate hydroxide hydrate synthesized by hydrothermal processes. QL-ZnO, QD-ZnO and QDL-ZnO are the precursors of L-, D- and DL-ZnO, respectively.

Comment 6:

Figure S7h: the crystal lattice fringe should be $2.64 \text{ nm} / 9 = 0.2933 \text{ nm}$ rather than *ca.* 0.26 nm. Please double-check.

Response 6:

Thanks for the very valuable comment. We have carefully checked the lattice fringe, and it was indeed *ca.* 0.26 nm. Actually, we incorrectly labeled the lattice spacing of 10 as 9 in original figure. We have corrected this mistake in the revised Figure S7h.

Modification 6:

1. The Figure S7h was revised.

Figure S7. TEM images of (a) L-ZnO, (b) D-ZnO and (c) DL-ZnO. HRTEM images and the corresponding lattice fringe spacing of (d) and (g) L-ZnO, (e) and (h) D-ZnO, (f) and (i) DL-ZnO.

Comment 7:

Figure S8b-b2: why a different scale (250 nm) was used for these figures? It would be better to use the same scale as shown in S8a and S8c (i.e., 200 nm).

Response 7:

Thanks for the very valuable suggestion. We uniformed the scale to 200 nm in revised Figure S8b-b2.

Modification 7:

1. Figure S8b-b2 was revised as follows.

Figure S8. Energy-dispersive X-ray spectroscopy (EDS) of (a) L-ZnO, (b) D-ZnO and (c) DL-ZnO.

Comment 8-10:

Figure S9: pore size distribution should be also provided in this figure.

Are there any mesopores or hierarchical porosity in the as-synthesized materials?

The authors mentioned that “all of the fabricated ZnO samples show a similar surface area (Figure S9, SI),” But they did not tell what the pore size is for the as-synthesized porous materials. In literature, the presence of mesopores (or macropores) favors multilight scattering/reflection, resulting in enhanced harvesting of the exciting light and thus improved photocatalytic activity (CrystEngCommun 2022, 24, 6498-6504; J. Clean. Prod. 2021, 328, 129745; Environ. Chem. Lett. 2021, 19, 3573-3582). The authors are suggested to consider similar effects from the mesopores (or macropores) if applicable.

Response 8-10:

Thanks for the very valuable suggestions. We have provided the pore size

distribution of samples in Figure S9b. As shown in Figure S9b, L-ZnO, D-ZnO and DL-ZnO all possess the similar mesopores with size of 2.8 nm, and they also have mesopores with size of 40.1, 29.8 and 18.3 nm, respectively. The total pore volumes are 0.18, 0.20, and 0.18 cm³/g, respectively. Although the presence of mesoporous structure is beneficial to enhance light scattering/reflection and provide active sites, the difference in between the three samples is very small, so it should be not the key factor to affect the PEC performance.

Modification 8-10:

1. Figure S9 was supplemented as follows.

Figure S9. (a) N₂ adsorption-desorption isotherm curves and (b) Pore size distribution of L-ZnO, D-ZnO and DL-ZnO.

2. The following sentences were supplemented in Supporting Information.

The surface area and porosity structure have important effects on the light absorption and activity of photocatalysts¹⁻³, which were investigated by N₂ adsorption-desorption measurements (Figure S9, SI). All samples show typical IV isotherms with H₃-type hysteresis loops, indicating the presence of mesopores. The BET surface area values of L-ZnO, D-ZnO and DL-ZnO are 30.3, 36.3 and 39.3 cm²/g, respectively. Moreover, L-ZnO, D-ZnO and DL-ZnO possess the similar mesopores with size of 2.8 nm, and they also have mesopores with size of 40.1, 29.8 and 18.3 nm, respectively. The total pore volumes of L-ZnO, D-ZnO and DL-ZnO are 0.18, 0.20, and 0.18 cm³/g, respectively. Although the presence of mesoporous structure is beneficial to enhance light scattering/reflection and provide active sites,

the photocatalytic activity difference between the three samples due only to the surface area and porosity structure will be relatively small.

3. The following references were supplemented in Supporting Information.

1. Wang, D. *et al.* Low-cost synthesis of a nanocomposite of MoS₂ and alkali-activated halloysite nanotubes for photocatalytic RhB degradation. *CrystEngComm* **24**, 6498-6504 (2022).
2. Liu, Y. *et al.* Synergistic effects of g-C₃N₄ three-dimensional inverse opals and Ag modification toward high-efficiency photocatalytic H₂ evolution. *J. Clean. Prod.* **328**, 129745 (2021).
3. Hao, M. *et al.* Higher photocatalytic removal of organic pollutants using pangolin-like composites made of 3-4 atomic layers of MoS₂ nanosheets deposited on tourmaline. *Environ. Chem. Lett.* **19**, 3573-3582 (2021).

4. Page 6, Lines 5-8, the following sentences were revised.

All ZnO samples show the similar surface area and pore structure distribution (Figure S9, SI), and the mesoporous structure can enhance the light scattering/reflection and provide abundant active sites⁴³.

5. The following references were supplemented.

43. Liu, Y. *et al.* Synergistic effects of g-C₃N₄ three-dimensional inverse opals and Ag modification toward high-efficiency photocatalytic H₂ evolution. *J. Clean. Prod.* **328**, 129745 (2021).

Comment 11:

The authors stated that “the high-resolution O 1s spectra in Figure S10b (SI) can be indexed into two distinctive binding energy peaks at 529.9 and 531.8 eV, which belong to the lattice oxygen and adsorbed oxygen/hydroxyl radicals on the surface, respectively”. Please provide literature here.

Response 11:

Thanks for the very valuable suggestion. The high-resolution O 1s spectra was re-fitted carefully, with two distinctive binding energy peaks at 530.5 and 532.1 eV, and we provided the supporting literatures in the revised manuscript.

Modification 11:

1. Page 6, Lines 9-12, the following sentences have been added:

The high-resolution O 1s spectra fitted in Figure S10b (SI) can be indexed into two distinctive binding energy peaks at 530.5 and 532.1 eV, which belong to the lattice oxygen and adsorbed oxygen species, respectively^{44, 45}.

2. The following references have been added:

44. Jiang, P. *et al.* Precursor engineering to reduce processing temperature of ZnO films for flexible organic solar cells. *Angew. Chem. Int. Ed.* **61**, e202208815 (2022).
45. Zhao, X. *et al.* Metal-organic framework-derived ZnO/ZnS heteronanostructures for efficient visible-light-driven photocatalytic hydrogen production. *Adv. Sci.* **5**, 1700590 (2018).

3. The fitting results in Figure S10 (SI) have been revised:

Figure S10. High-resolution XPS of (a) Zn 2p and (b) O1s of L-ZnO, D-ZnO and DL-ZnO.

Comment 12:

Reference 14: volume number (32) is missing. In addition, the article number is 2003297 rather than e2003297.

Response 12:

Thanks very much for pointing out this error. We have checked our manuscript carefully and corrected the mistakes in the revised manuscript.

Modification 12:

15. Sun, Y. *et al.* Spin-related electron transfer and orbital interactions in oxygen electrocatalysis. *Adv. Mater. Interfaces* **32**, 2003297 (2020).

Comment 13:

Methods: please change “1 mL o-tolidine indicator solution” to “1 mL of o-tolidine indicator solution”; change “20 mg catalyst decorated with CoO_x cocatalyst” to “20 mg of catalyst decorated with CoO_x cocatalyst”; change “0.01 mol·L⁻¹” to “0.01 mol L⁻¹”. Please correct similar errors throughout the manuscript.

Response 13:

Thanks for the very valuable suggestion. We have checked the whole manuscript and corrected them in the revised manuscript.

Modification 13:

1. Page 16, Lines 21-23, the following sentences have been revised:

After that, 3 mL of the electrolyte solution was mixed with 1 mL of *o*-tolidine indicator solution, and the absorption spectra were monitored using a Shimadzu UV-2600 spectrometer.

2. Page 16, Lines 25-26, the following sentences have been revised:

20 mg of catalyst decorated with CoO_x cocatalyst was dispersed in H₂O (100 mL) with AgNO₃ (0.01 mol L⁻¹) as sacrificing agent.

3. Page 16, Lines 10-12, the following sentences have been revised:

For RRDE measurements, 5 mg of ZnO powder that was scraped off from ZnO photoanodes was dispersed into 625 μL of water, 625 μL of ethanol and 25 μL Nafion solution (5 wt%) under sonication for 60 min.

Comment 14:

Reference 17: article number is 1907976 rather than e1907976.

Response 14:

Thanks for the very valuable suggestion. We have revised the reference.

Modification 1:

18. Chen, R. R. *et al.* Antiferromagnetic inverse spinel oxide LiCoVO₄ with spin-polarized channels for water oxidation. *Adv. Mater.* **32**, 1907976 (2020).

Comment 15:

Reference 23: volume number (6) is missing.

Response 15:

Thanks for the very valuable suggestion. We have added the volume numbers (6) to reference 23.

Modification 15:

24. Gao, W. *et al.* Electron spin polarization-enhanced photoinduced charge separation in ferromagnetic ZnFe₂O₄. *ACS Energy Lett.* **6**, 2129-2137 (2021).

Comment 16:

It would be better to compare the photocatalytic performance of the as-synthesized photocatalysts in this study with that reported in the literature to justify that this study significantly advances the development in this field and thus merits publication.

Response 16:

Thanks for the very valuable suggestion. We compared the photocurrent of chiral ZnO with those of ZnO-based photoanodes in previous literatures. And the chiral ZnO shows much better performance, confirming the advance of our strategy to improve the photocatalytic performance.

Modification 16:

1. Table S1 has been added in Supporting Information.

Table S1. Performance comparison of ZnO-based photoanodes in this work and previous literatures.

Photoanode	Modification strategy	Electrolyte	Photocurrent	Reference
L-ZnO	Chiral structure	0.5 M Na ₂ SO ₄	0.43 mA/cm ² at 1.23 V _{RHE}	This work

D-ZnO			0.40 mA/cm ² at 1.23 V _{RHE}	
ZnO (002)	Facet engineering	0.5 M Na ₂ SO ₄	0.35 ± 0.01 mA/cm ² at 1.2 V _{RHE}	4
ZnO/g-C _x N _y (1.0)	Conformal coating	0.1 M KOH	~0.25 mA/cm ² at 1.23 V _{RHE}	5
ZnO:Co@ZIF-8	Doping and functionalization	pH=13 alkaline solution	<0.16 mA/cm ² at 1.33 V _{RHE}	6
15 nm SnS ₂ /b-ZnO NW	Heterojunction	0.35 M Na ₂ SO ₃ + 0.25 M Na ₂ S	0.36 mA/cm ² at 1.23 V _{RHE}	7
In doped ZnO	Element doping	0.1 M Na ₂ SO ₄	0.42 mA/cm ² at 1.23 V _{RHE}	8
Al doped ZnO	Element doping	0.5 M Na ₂ SO ₄	0.31 mA/cm ² at 1.23 V _{RHE}	9
C-ZnO NRs			14.8 μA/cm ² at 0.9 V _{RHE}	
N-ZnO	Element doping	0.5 M Na ₂ SO ₄	13 μA/cm ² at 0.9 V _{RHE}	10
S-ZnO			6 μA/cm ² at 0.9 V _{RHE}	
ZnO/ZnS	Heterostructure	0.5 M Na ₂ SO ₃	0.073 mA/cm ² at 0.78 V _{RHE}	11
ZnO@ZIF-8/67	Ternary hierarchical semiconductor	0.5 M Na ₂ SO ₄	0.11 mA/cm ² at 1.23 V _{RHE}	12
ZnO QDs/NRs	Homojunction	0.5 M Na ₂ SO ₄	0.42 mA/cm ² at 1.23 V _{RHE}	13
ZnO@N-CD@ZIF- 8	Photosensitization	0.5 M Na ₂ SO ₄	~0.30 mA/cm ² at 0.73 V _{RHE}	14

2. The related references cited by Table S1 have been supplemented in Supporting Information.

4. Pawar, A. U. *et al.* Crystal facet engineering of ZnO photoanode for the higher water splitting efficiency with proton transferable nafion film. *Nano Energy* **20**, 156-167 (2016).

5. Hajduk, Š. *et al.* Conformal carbon nitride coating as an efficient hole extraction layer for ZnO nanowires-based photoelectrochemical cells. *Adv. Mater. Interfaces* **4**, 1700924 (2017).
6. Galan-Gonzalez, A. *et al.* Cobalt-doped ZnO nanorods coated with nanoscale metal-organic framework shells for water-splitting photoanodes. *ACS Appl. Nano Mater.* **3**, 7781-7788 (2020).
7. Bagal, I. V. *et al.* Investigation of charge carrier dynamics in beaded ZnO nanowire decorated with SnS₂/IrO_x cocatalysts for enhanced photoelectrochemical water splitting. *Appl. Surf. Sci.* **613**, 156091 (2023).
8. Le, N. *et al.* Enhanced light absorption and charge separation of In-doped ZnO nanorod arrays for photoelectrochemical water-splitting application. *Int. J. Energ. Res.* **46**, 6264-6276 (2021).
9. Kant, R. *et al.* Fabrication of ZnO nanostructures using Al doped ZnO (AZO) templates for application in photoelectrochemical water splitting. *Appl. Surf. Sci.* **447**, 200-212 (2018).
10. Karmakar, K. *et al.* Stable and enhanced visible-light water electrolysis using C, N, and S surface functionalized ZnO nanorod photoanodes: engineering the absorption and electronic structure. *ACS Sustainable Chem. Eng.* **4**, 5693-5702 (2016).
11. Li, C. *et al.* ZnO/ZnS heterostructures grown on Zn foil substrate by hydrothermal method for photoelectrochemical water splitting. *Int. J. Hydrogen Energy* **44**, 25416-25427 (2019).
12. Jia, G. *et al.* 1D alignment of ZnO@ZIF-8/67 nanorod arrays for visible-light-driven photoelectrochemical water splitting. *Appl. Surf. Sci.* **448**, 254-260 (2018).
13. Chen, Y.-C. *et al.* Overall photoelectrochemical water splitting at low applied potential over ZnO quantum dots/nanorods homojunction. *Chem. Eng. J.* **368**, 746-753 (2019).
14. Han, H. *et al.* Highly ordered N-doped carbon dots photosensitizer on metal-organic framework-decorated ZnO nanotubes for improved

photoelectrochemical water splitting. *Small* **15**, 1902771 (2019).

3. Page 11, Lines 4-6, the following sentences have been added:

In addition, the performance improvement is better than the reported modification strategies for ZnO-based photoanodes (Table S1, SI).

Note: All the suggestions have been incorporated and the manuscript has been revised carefully for English language errors. We hope our revised manuscript could meet the publication requirement.

Sincerely yours,

Prof. Ji-Jun Zou

Tianjin University, China.

REVIEWERS' COMMENTS

Reviewer #2 (Remarks to the Author):

The authors responded properly and in details to all comments and hence I recommend publication.